Recapitulating human ovarian aging using random walks

http://orcid.org/0000-0002-6016-8089 Johnson Joshua 1 joshua.2.johnson@cuanschutz.edu
Emerson John W. 2
Lawley Sean D. 3
1 Department of Obstetrics and Gynecology, University of Colorado-Anschutz Medical Center , Aurora, Colorado , United States
2 Department of Statistics and Data Science, Yale University , New Haven, Connecticut , United States
3 Department of Mathematics, University of Utah , Salt Lake City, Utah , United States
Uversky Vladimir
Electronic publication date: 2022 Aug 22
Publication date: 2022
Volume: 10
Electronic Location ID: e13941
Received 2022 Jun 9; Accepted 2022 Aug 2
Copyright: © 2022 Johnson et al.
Copyright year: 2022
Copyright holder: Johnson et al.
License: This is an open access article distributed under the terms of the Creative Commons Attribution License, which permits unrestricted use, distribution, reproduction and adaptation in any medium and for any purpose provided that it is properly attributed. For attribution, the original author(s), title, publication source (PeerJ) and either DOI or URL of the article must be cited.
License URL: https://creativecommons.org/licenses/by/4.0/

Keywords: Aging, ANM, DNA damage, EIF2S1, Follicle, Fertility, Ovary, Mathematical modeling, Menopause, Primordial follicle

Funding: CU-Anschutz Department of Obstetrics and Gynecology Research Funds McPherson Family Funds NSF CAREER DMS-1944574 NSF DMS-1814832 Joshua Johnson is supported by CU-Anschutz Department of Obstetrics and Gynecology Research Funds and McPherson Family Funds. Sean D. Lawley is supported by NSF CAREER DMS-1944574 and NSF DMS-1814832. The funders had no role in study design, data collection and analysis, decision to publish, or preparation of the manuscript.

==============================
Mechanism(s) that control whether individual human primordial ovarian follicles (PFs) remain dormant, or begin to grow, are all but unknown. One of our groups has recently shown that activation of the Integrated Stress Response (ISR) pathway can slow follicular granulosa cell proliferation by activating cell cycle checkpoints. Those data suggest that the ISR is active and fluctuates according to local conditions in dormant PFs. Because cell cycle entry of (pre)granulosa cells is required for PF growth activation (PFGA), we propose that rare ISR checkpoint resolution allows individual PFs to begin to grow. Fluctuating ISR activity within individual PFs can be described by a random process. In this article, we model ISR activity of individual PFs by one-dimensional random walks (RWs) and monitor the rate at which simulated checkpoint resolution and thus PFGA threshold crossing occurs. We show that the simultaneous recapitulation of (i) the loss of PFs over time within simulated subjects, and (ii) the timing of PF depletion in populations of simulated subjects equivalent to the distribution of the human age of natural menopause can be produced using this approach. In the RW model, the probability that individual PFs grow is influenced by regionally fluctuating conditions, that over time manifests in the known pattern of PFGA. Considered at the level of the ovary, randomness appears to be a key, purposeful feature of human ovarian aging.

Introduction

Human ovarian aging depends upon the rate of loss of a dormant reserve of primordial follicles (PFs) to a growth phase (Richardson, Senikas & Nelson, 1987; Erickson, 2008; Wallace & Kelsey, 2010). Individual PFs consist of a single immature egg cell (oocyte) surrounded by a layer of non-proliferative somatic pregranulosa cells. PF growth activation (PFGA) consists of pregranulosa cells entering into a slow but active cell cycle, and they are now termed granulosa cells (Kallen, Polotsky & Johnson, 2018). PF commitment to growth and therefore loss from the reserve (also referred to as “decay”) is thought to be irreversible.

Follicle numbers across the lifespan from the time of their development during fetal life (Pepling & Spradling, 1998, 2001) through postmenopausal life have been assessed directly in histological preparations (Wallace & Kelsey, 2010; validated by McLaughlin et al. (2015, 2017)). Of the hundreds of thousands of PFs in the human PF reserve, most are destined to die some time after PFGA in a process termed atresia, and only a very small fraction survives to ovulate a mature egg once per menstrual cycle. Human menopause occurs when the number of arrested PFs drops below a threshold of hundreds to perhaps a thousand (Richardson, Senikas & Nelson, 1987; Faddy & Gosden, 1996; De Vos, Devroey & Fauser, 2010; Santoro & Johnson, 2019; Ford et al., 2020). At this time, the supply of growing follicles essentially ceases, leading to the loss of menstrual cyclicity. The age at natural menopause (ANM) is such that approximately 1 in 250 women reach it at or before age 35, and 1 in 100 at or before age 40. The median ANM is 51, and very few women reach menopause after age 62 (Gold, 2011). A central goal in reproductive biology and medicine is the determination of mechanisms that dictate the decision of individual PFs to undergo PFGA, and how this results in the patterns of decline seen in individual women (e.g., “subjects”) in order to give rise to the known ANM distribution.

Cellular stress and particularly oxidative damage have long been associated with ovarian aging (Agarwal, Gupta & Sharma, 2005; Tatone et al., 2006, 2008; Oktay et al., 2020), and one of our groups is probing how endogenous physiological stress and damage impact PFs directly. Using a combination of wet laboratory and bioinformatics approaches, we have recently identified the stress-and-damage-regulated Integrated Stress Response (ISR) pathway (Pakos-Zebrucka et al., 2016; Costa-Mattioli & Walter, 2020) as a new potential key regulator of ovarian aging throughout the ovary and at the level of individual PFs (Llerena Cari et al., 2021). In that work we showed that the PFGA stimulator Tnfα leads to a rapid ISR response, including the upregulation of the translation of proteins involved in cellular repair. As its name suggests, the ISR coordinates responses to a variety of cellular insults, integrating them so that conserved cellular machinery can respond to and repair induced damage. Successful cellular repair can result in ISR checkpoint resolution and a return to the active cell cycle. Based upon this, we have developed a model (summarized in the Graphical Abstract, Fig. 1) based upon physiological regional fluctuations (Fig. 1A) of ISR activity that occur in granulosa cells during normal ovarian function (Llerena Cari et al., 2021). We hypothesize that it is checkpoint resolution of physiological stress and DNA damage that allows a switch to an active cell cycle and PFGA (Kaufmann & Paules, 1996; Shaltiel et al., 2015; Bartek & Lukas, 2007; Marini et al., 2006; Smits & Gillespie, 2015).

Figure 1 Graphical abstract: modeling primordial ovarian follicle growth activation with random walks.

The control mechanism that determines when individual primordial follicles (PFs) begin to grow over time is unknown. The ovary is a uniquely dynamic organ that undergoes constant remodeling (ovaries, A) as follicles grow and die, and sometimes ovulate and form corpora lutea. The organ also changes with age, and this includes diminishing numbers of follicles, alterations in the ovarian stroma, and changes in blood vessels and their distribution. These dynamic changes include regional signaling differences in the ovary (intra-ovarian “microenvironments”), including factors that induce stress and damage over time (grey arrow). In (A), the same dormant PF (red arrowhead) and its immediately surrounding ovarian region are monitored over time. As structures develop, change, and die in that region, the monitored PF is subject to dynamically changing signals and physiological damage-inducing agents that activate the ISR. Spatiotemporally fluctuating ISR activity is simplified in (B) (modified from Llerena Cari et al. (2021)). When ISR activity is high, CELL CYCLE ARREST and CYTOPROTECTION occur due to checkpoint activation. If ISR activity declines enough in a PF to the point of ISR checkpoint resolution, a Growth Threshold is crossed, and pregranulosa cell cycle entry and GROWTH occurs (asterisk, monitored PF). In (B), three PFs experience fluctuating ISR activity due to fluctuating stress and damage (y-axis) over time (x-axis), and this is modeled as random walks (RWs). Each PF begins the RW at time 0 (“birth”, PF on y-axis), and dark plot lines indicate changing ISR activity as a RW over time (x-axis). Two PFs cross the PFGA threshold at approximately ages 21 and 38, and the monitored PF in (A) crosses the threshold at age 52 (red triangle, asterisk). Impacts of the following potential sources of variability between simulated women upon RW patterns and the timing of PF exhaustion are considered. First, variation in PF numbers between subjects around the time of birth is modeled according to the reported distribution (starting supply, Wallace & Kelsey (2010)). Second, the probability of PF movement is modified so that “drift” towards the growth threshold occurs (B, orange block arrow at right); the amount of drift is optionally modeled to vary between simulated subjects.

Quantitative descriptions of the pattern of loss of PFs from the ovarian reserve have a long history (Faddy, Jones & Edwards, 1976; Gougeon, Ecochard & Thalabard, 1994; Faddy & Gosden, 1996; Hansen et al., 2008; Faddy, 2000; Coxworth & Hawkes, 2010). Histological specimen-derived numbers of PFs across the lifespan were fit using a variety of types of equations (i.e., power, differential equation, biphasic, etc.). In one report, the accuracy of fit of different equations was evaluated, and a differential equation model was found to generate a curve that best matched human PF decay (Coxworth & Hawkes, 2010). In a separate simulation-based approach, we evaluated probabilities of PFGA over time that can result in PF reserve loss that can match patterns seen in nature (bioRxiv preprint, Johnson et al., 2016). While these approaches described follicle loss quite accurately, and can be interpreted post hoc in terms of how known regulators of PFGA are likely to function, none of them can be considered a mechanistic “forward” derivation of patterns of follicle loss.

We reasoned that if the biological mechanisms at work in individual PFs that dictate the probabilities of arrest or growth over time can be clarified, we might be able to simulate PFGA over time in a way that recapitulates the natural pattern. Our ISR data and proposed model (Llerena Cari et al., 2021) suggested that individual PFs experiencing regionally fluctuating stress and damage (and thus ISR activity) over time might reflect a situation analogous to a random walk (RW; Berg, 1993; Billingsley, 1995; Codling, Plank & Benhamou, 2008) relative to a threshold of growth activation (Fig. 1B). If so, threshold crossing would occur randomly for individual PFs over (simulated) time. RW models consist of simulated conditions that exhibit change over time such that a positional value of X varies by “walking,” taking steps of size Δx, and the direction of each step is random (Codling, Plank & Benhamou, 2008). In continuous RWs, the size of stochastic fluctuations is referred to as “diffusivity” (variable D), and deterministic (i.e., a degree of nonrandom) motion is referred to as “drift” (V). When drift is present in a one-dimensional RW with two possible directions of movement, the probability of movement in either direction is not equal. We hypothesized that if all PFs within the ovary are considered in terms of the local signals they receive and a threshold of PFGA exists, modeling PFs as undergoing one-dimensional RWs relative to that threshold would recapitulate their loss to growth over time. To our knowledge, our approach is the first to prospectively model PFGA and ovarian aging as a manifestation of a random process, the RW.

In this study, we used RWs (Norris, 1998) to model the behavior of the PF reserve and monitored patterns of ovarian aging in simulated subjects and in populations of simulated women. Formal mathematical analysis was used to produce continuous RWs as seen when determined by Brownian motion/diffusion (Gardiner, 2009). A corresponding discrete step (e.g., non-continuous) RW model was developed using the R statistical programming language (R Core Team, 2021), and conditions were established where PFs executed RW steps representing fluctuating ISR activity over the discrete simulated months of the human postnatal lifespan.

Different approaches were used to model plausible variation between subjects, and to evaluate its impact on simulated ovarian aging. These included (i) the impact of the known distribution of initial PF number (a subject’s “starting supply” Depmann et al. (2015)), (ii) homogenous vs. heterogeneous ISR action (modeled as “drift” V within the RW, graphical abstract, double-headed arrow) in simulated subjects within populations and (iii) time-invariant vs. time-variant drift in simulated subjects within populations. In all cases, model output was compared to a benchmark dataset of actual PF numbers reflective of decay over time (Wallace & Kelsey, 2010) and the known distribution of the human ANM. Annotated code for re-analysis and reproduction of output is provided. When ISR activity within PFs is simulated using RWs that occur relative to a threshold, output can be seen to closely match natural patterns of ovarian aging in subjects and in population(s) of women.

Methods

Data sourcing and definitions

We performed a literature review of reports of numbers of primordial/nongrowing follicles, and established parameters for our modeling approach based upon (i) the distribution(s) of measured numbers of PFs present in the human ovary over time (Wallace & Kelsey, 2010) and (ii) the degree of variability between women in terms of their cessation of ovarian function (Weinstein et al., 2003; Nelson, 2009; Simpson, 2008; Persani, Rossetti & Cacciatore, 2010; McKinlay, Bifano & McKinlay, 1985; McKinlay, Brambilla & Posner, 1992). ANM data used in this study were generated from published plots (McKinlay, Bifano & McKinlay, 1985; McKinlay, Brambilla & Posner, 1992) using the data extraction tool WebPlotDigitizer (Rohatgi, 2021). Although loss of ovarian function prior to the age of 40 is defined clinically as reflective of a pathological state, termed primary ovarian insufficiency (POI; Nelson (2009)), we included the possibility that random action could result in measurable numbers of women that exhaust their PF reserve by that time.

Matlab and R code

Data were downloaded from their respective sources and analyzed using Matlab or R (R Core Team, 2021) as indicated in order to interrogate RW modeling of datasets. Matlab (https://doi.org/10.6084/m9.figshare.19834774.v1) and R (https://doi.org/10.6084/m9.figshare.19858987.v1) code used in the manuscript are available in a public repository for download and use.

Results

Our identification of the ISR as a potential regulatory mechanism rests upon the concept that ISR activity fluctuates over time within PFs due to physiological processes that fluctuate over time regionally within the ovary (Fig. 1; data evaluating mouse PFs in Llerena Cari et al., 2021). We hypothesized that modeling ISR activity as a one-dimensional RW would generate patterns of follicle growth activation (and thus loss from the ovarian reserve) if that RW included a threshold for the state change between dormancy and growth.

Mathematical model of fluctuating ISR activity and PFGA

For a single PF in a given woman, we model fluctuating ISR activity by a RW. PFGA occurs when the ISR activity of this PF crosses a threshold. Crossing that threshold results in a PF’s subtraction from the ovarian reserve.

To describe our mathematical model precisely, let X(t) denote the ISR activity of a single PF at time t ≥ 0. Here, time t is the age of the woman so that time t = 0 corresponds to her birth.

To model ISR fluctuations as in Llerena Cari et al. (2021) (see also Fig. 1), suppose that the ISR activity in this PF either increases or decreases by an amount Δx > 0 over a time step Δt > 0. Mathematically, this is expressed as

(1) X((n+1)Δt)={X(nΔt)+Δxwithprobabilityp,X(nΔt)−Δxwithprobability1−p,

where p is the probability that ISR activity increases. The lefthand side of Eq. (1) is the ISR activity after n + 1 time steps, which is given by the ISR activity after n time steps (i.e., X(n Δt)) plus or minus the amount Δx. This type of model is called a random walk because the value of X “walks” by taking steps of size Δx, and the direction of each step (either up or down) is random (Berg, 1993).

If the steps Δt and Δx are small, then the discrete random walk in Eq. (1) is equivalent to the continuous random walk whose dynamics are described by the stochastic differential equation (Gardiner, 2009),

(2) dX=−Vdt+2DdW,

where W(t) is a standard Brownian motion and

(3) D=(Δx)22Δt,V=2ΔxΔt(12−p).

The equivalence of the discrete model Eq. (1) and the continuous model Eq. (2) for small steps Δt and Δx is shown in the Appendix (Supplementary Information).

In words, Eq. (2) means that over an infinitesimal time dt, the ISR activity X changes by an amount dX equal to a deterministic amount −V dt plus a stochastic fluctuation 2DdW, where dW is normally distributed with mean zero and variance dt. The parameter D is called the “diffusivity” as it describes the size of the stochastic fluctuations, and V is called the “drift” as it describes the deterministic (i.e., nonrandom) motion. Biologically, the drift V represents the efficiency of cellular repair in our model. Notice that V > 0 if p < 1/2, which means that X tends to decrease.

We suppose that a PF begins to grow when its ISR activity drops below some “growth” threshold. We also suppose that a PF dies before it begins to grow if its ISR activity rises above some “death” threshold. Without loss of generality, we take the growth threshold to be at X = 0 and the death threshold at X = L > 0. Hence, a PF leaves the reserve at the first time τ such that X(τ) ∉ (0, L). Since paths of X are random, this reserve exit time τ is random.

Let N be the number of PFs in a given woman’s reserve at birth, which we refer to as her starting supply. Let F(t) denotes the number of PFs in a given woman’s reserve at time t (and thus F(0) = N). We assume that the reserve exit times for each of the N PFs are independent and identically distributed, and thus the expected number of PFs in the reserve at time t is

(4) E[F(t)]=NP(τ>t),

where E denotes averaging over the reserve exit times.

Random walk model recapitulates PF decay

PF Data from −0.25 years to 0.1 years in Wallace & Kelsey (2010) were selected in order to establish the starting supply distribution (Fig. S1). The median starting supply was found to be

(5) N=3.23×105.

For this median starting supply value Eq. (5), we sought parameter values for the random walk model to make the expected PF decay curve in Eq. (4) fit the PF counts reported by Wallace & Kelsey (2010). Since we can always rescale the ISR activity, without loss of generality we set the initial ISR activity to unity, X(0) = 1. This parameter search then yielded the following values for the diffusivity and drift,

(6) D=0.004year−1,V=0.051year−1,

and any value of the death threshold L ≥ 2. By taking the limit of a large death threshold (i.e., L→∞), we obtain the following formula for the expected follicle decay curve,

(7) E[F(t)]=N2[1+erf(1−Vt4Dt)−eV/D(1−erf(1+Vt4Dt))],

where erf(z):=2π∫0ze−y2dy is the Gauss error function. In the Appendix (Supplementary Information), we derive Eq. (7) and show that the difference between Eq. (7) and the expected follicle decay curve for any L ≥ 2 is negligible for D and V in Eq. (6).

The formula Eq. (7) with the median starting supply value N in Eq. (5) and D and V in Eq. (6) yields the solid blue curve in Fig. 2A. In particular, by tuning only two parameters (D and V in Eq. (6)), this mathematical mathematical model yields a follicle decay curve in Eq. (7) that closely fits the human follicle decay data reported by Wallace & Kelsey (2010).

Figure 2 Continuous one-dimensional random walk modeling of ovarian aging over simulated time.

(A) The number of PFs in ovarian histological preparations over more than six decades (red circles, data from Wallace & Kelsey, 2010). The overlaid blue solid line is our output from a RW model when the starting number of PFs is set to the population median. These RW settings result in the decay curve crossing an “ANM” threshold of 1,000 remaining PFs at 51 years (black horizontal lines in A and B). Additional decay curves were determined by applying different starting points from the PF starting supply distribution at time 0 from the same dataset (A, dashed blue lines, see also Fig. S1). Figure 2A depicts the outcome when homogenous drift is applied to simulated subjects (Subject-Identical drift). RW model output when drift was heterogeneous between subjects is shown in (B). In (A) and (B), dashed blue lines are RW output when starting supply was set arbitrarily to the values shown. Note that unlike homogenous drift between subjects where parallel decay lines result (A), heterogeneous drift results in decay that differs in trajectory between subjects, as seen in crossing blue dashed lines in B. In (C), ANM distributions reported by Weinstein et al. (2003) (blue diamonds) and (McKinlay, Bifano & McKinlay, 1985; McKinlay, Brambilla & Posner, 1992) (black squares) are overlaid with simulation output in the CDF plot. Red dots in (C) reflect the proportions of the population that are expected to reach menopause by the corresponding ages (e.g., approximately 1% of women reaching menopause before age 40, a median ANM of 51, and very few to no women reaching menopause after age 62). The blue dashed line in (C) is our RW output when only PF starting supply varies (e.g., homogenous drift), and the black dotted line in (C) is our RW output when drift is heterogeneous. Simulation output from these conditions are provided as histograms in Fig. S2 as compiled results from 10,000 simulations of the time that the 1,000-PF threshold was crossed by each subject.

In fact, the model curve in Eq. (7) fits the PF data nearly as well as the non-mechanistic curve used in Wallace & Kelsey (2010). More precisely, Wallace & Kelsey (2010) posited a flexible functional form (the so-called asymmetric double-Gaussian cumulative (ADC) curve), and then chose five free parameters in this phenomenological function so that the resulting curve fit this same PF count data, plus some prenatal PF counts. Here, “fit” is defined in terms of the sum of squared errors between the curve and the logarithm of the PF counts. By this measure of fit, our model curve in Eq. (7) is only 3% worse than this prior curve.

Random walk model recapitulates human ANM distribution

Human menopause occurs when the number of PFs in the ovarian reserve drops below a threshold of approximately 1,000 (Faddy & Gosden, 1995). Hence, we can use our mathematical model of PF decay to study menopause timing.

The solid blue PF decay curve in Fig. 2A crosses the 1,000 PF threshold (horizontal black line) at age 51 years. Hence, this curve corresponds to an ANM equal to 51 years. However, there is considerable population variability in the starting supply N of PFs at birth. All else being equal, women with starting supplies higher (respectively, lower) than the median starting supply will tend to reach menopause later (respectively, earlier) than age 51.

To understand how starting supply population variability translates into ANM population variability, we first characterize the starting supply population distribution. As our starting supply data, we use the 30 PF counts in Wallace & Kelsey (2010) taken from women within a few months of birth.

We show in the Appendix (Supplementary Information) that these data are well-described by a log-normal distribution with parameters

(8) μ=12.686,σ=0.497.

That is, we model the distribution of the starting supply as

(9) N=exp⁡(μ+σZ),

where Z is a standard normal random variable and μ and σ are in Eq. (8). The median of this starting supply distribution in Eq. (9) is exp(μ) = 3.23 × 105 given in Eq. (5). As pointed out above, the solid blue PF decay curve in Fig. 2A that starts from this median starting supply yields an ANM equal to 51 years. Interestingly, the median ANM across a population is also approximately 51 years (Weinstein et al., 2003; McKinlay, Bifano & McKinlay, 1985; McKinlay, Brambilla & Posner, 1992). Hence, this solid blue curve in Fig. 2A can be understood as describing the “median woman.” The dashed blue curves in Fig. 2A show the PF decay curves for different quantiles of the starting supply distribution in Eq. (9) (namely, the 1%, 5%, 25%, 75%, 95%, 99% quantiles).

Model ANM output is compared with available information about the human ANM distribution as follows. First, the ANM distribution across a population of women who only vary in their starting supply N (as in Fig. 1A) is considered. In particular, the blue dashed curve in Fig. 2C was generated by simulating the PF decay dynamics of 104 women, where each woman begins with a starting supply N sampled independently from the log-normal distribution in Eq. (9), and the ANM for each of the 104 simulated women is the time when their PF reserve drops below 1,000. The PF decay dynamics for each woman follow the RW model described above, with diffusivity D and drift V in Eq. (6). The blue diamonds in Fig. 2C is the ANM distribution reported by Weinstein et al. (2003) when only variable starting supply was present in the RW model and drift was fixed, model output (blue dashed line) and the empirical (Weinstein et al., 2003). ANM distribution (blue diamonds) were in close agreement. We emphasize that this agreement follows merely from combining the empirical starting supply distribution in Eq. (9) with our RW model, where the two free parameters in the RW model (D and V in Eq. (6)) were chosen to fit PF decay data.

The agreement between the model and the ANM data in Fig. 2C is compelling, but there are some caveats. First, the ANM data from McKinlay, Brambilla & Posner (1992) in Fig. 2C (black squares) is evidently more variable than the ANM data from Weinstein et al. (2003), (blue diamonds). Second, the only source of population variability in the model for the blue curve in Fig. 2C is in starting supply. That is, each of the 104 simulated women has identical diffusivity and drift parameters D and V in Eq. (6). However, it is not plausible that all intrinsic (e.g., genetic, epigenetic) and extrinsic (environmental) conditions that determine PFGA timing are entirely identical between women.

To address these two issues, we introduced an additional source of population heterogeneity into our model by allowing the drift parameter V to vary between simulated women in addition to the starting supply N. There are many possible choices that we could make for the population distribution of V, but for concreteness and we allow V to vary between women according to

(10) V=V¯(1+cY),

where V¯=0.051year−1 as in Eq. (6), c = 0.03, and Y is a standard normal random variable independent of N in Eq. (9). In words, Eq. (10) simply means that the drift parameter V for each woman is normally distributed with mean (and median) V¯ and 3% coefficient of variation.

The solid blue PF decay curve in Fig. 2B is identical to the “median woman” solid blue curve in Fig. 2A, since it is for the median starting supply in Eq. (5) and the median of the drift distribution in Eq. (10). The dashed blue curves in Fig. 2B are for the same quantiles of the starting supply distribution as in Fig. 2A, but the drift parameters for each of these six curves are sampled from the distribution in Eq. (10). Notice that these dashed curves sometimes cross, which means that a woman with a large starting supply may at some point have less PFs in her reserve than a woman with a smaller starting supply, due to differences in their drift parameters. Biologically, this represents individual differences in the rate of follicle loss over time as might be influenced by genetics or environmental exposures.

The black dotted curve in Fig. 2C shows the ANM distribution resulting from a variable starting supply and a variable drift. This ANM distribution was generated from the model analogous to the blue curve in Fig. 2C, except the drift parameter for each simulated woman varied according to Eq. (10). This plot shows that by introducing population heterogeneity into the drift parameter, the model yields an ANM distribution in line with the McKinlay, Brambilla & Posner (1992) data. Histograms depicting the ANM distributions produced by each of the RW conditions are shown in Fig. S2.

RW model flexibility

The analysis above shows that a simple RW model can recapitulate some prominent features of ovarian aging seen in nature. However, ovarian aging is a complex, multi-faceted process involving a variety of dynamic components. The purpose of this section is to show how our RW framework can be adapted to model different aspects of ovarian biology.

For example, the RW modeling framework can be used to investigate the effects of an acute drop in PF number. Such acute drops are common effects of certain cancer treatments (Fleischer, Vollenhoven & Weston, 2011; Das et al., 2012; Dunlop & Anderson, 2015; Chan et al., 2015). In Fig. 3A, RW traces for 50 simulated subjects are shown when Subject-Variable drift (identical to that shown in Fig. 2B) is applied, but PF starting supply is fixed at the reported median. The impact of an acute drop in PF number at approximately simulated age 12 is shown (asterisk, after De Vos, Devroey & Fauser (2010)). After PF number is adjusted in this way, RW traces are again shown for 50 simulated subjects, and an acceleration in the times that the ANM threshold is crossed is apparent when compared to uninterrupted decay. The RW modeling framework can also be used to investigate the purported acceleration of PF loss during human ovarian aging (Faddy et al., 1992) thought to arise due to declining levels of Antimullerian hormone (AMH) late in the third decade. While definitive experimental evidence that AMH slows PFGA and ovarian aging in other mammals is available (Durlinger et al., 1999; Pankhurst, 2017), it is less clear that AMH has this same role during ovarian aging in women in vivo (Leidy, Godfrey & Sutherland, 1998).

Figure 3 Further exploration of primordial follicle loss with aging: ANM variability, impact of simulated acute loss, and simulation of accelerating growth activation with time.

In (A), identical conditions were used as in as in Fig. 2B with subject-variable drift, but here, results from 50 RWs that all begin at the median starting supply value are traced. ANM variability given the same starting supply but varying drift is shown in the varied points that dashed decay curves cross the ANM threshold (black horizontal line; see also histogram in Fig. S2). Also in a, 50 resulting trajectories from a simulated subject that experienced an acute loss of PFs at approximately 12 years are shown (indicated by *). Next, in distinct simulation conditions, the drift applied was constant until age 38 years, when a one-time elevation in drift is applied for the remainder of the simulation. The resulting ANM distribution from 10,000 subjects under the conditions used in (B) is provided as a histogram in Fig. S2.

Acceleration in PF loss as might be seen in response to declining AMH levels can be incorporated into the model by allowing the drift parameter to increase over time. To illustrate, we let the mean drift parameter in Eq. (10) change at age 38 according to

(11) V¯=(0.024year−1beforeage38,0.033year−1afterage38.

Given that the acceleration in PF loss around this time is more likely to be continuous, as the decline in AMH levels can be seen to be continuous in this window of time (Kelsey et al., 2011), we also tested more continuous drift modifications, but the difference between those and the simple stepwise drift modification was negligible (not shown). Figure 3B shows example PF decay trajectories for women with different starting supplies (equal to the quantiles in Fig. 2A) and whose drift parameters varied in time according to Eq. (11) and varied between women according to Eq. (10).

When Subject-Variable drift was applied and stepwise acceleration occurred (Fig. 3B), an ANM distribution was produced that again greatly resembles the actual ANM distribution (Fig. S2). In addition to a direct visualization of the impact of accelerating drift as might be expected due to declining AMH levels as women approach their forties, this approach demonstrates the flexibility of the RW model as it can be modified in order to begin to address these questions.

Software for further investigation

As illustrated above, a variety of biological factors can be investigated using the RW modeling framework. We therefore developed user-friendly R code for RW simulations with detailed documentation so that we ourselves and other interested parties can modify conditions, and, can add additional conditions that might influence PF loss (code available in a public repository, see the Methods section). This code allows one to alter a variety of conditions along possible degrees of freedom (e.g., variables like starting supply, acute follicle loss as shown in Fig. 3A, drift acceleration as in Fig. 3B, etc.) to investigate how such variables impact ovarian aging. Users can modify these conditions and add additional variables, perhaps as informed by their own experimental results.

Discussion

The ISR pathway responds to cellular damage and stress by upregulating the translation of factors that can repair damage, possibly allowing checkpoint resolution and the resumption of growth. Our prior work demonstrates that widespread and constitutive activation of the ISR in ovarian follicles contributes to blocked or very slow growth due to checkpoint activation. In the case of PFs, only those whose pregranulosa cells are able to achieve ISR checkpoint resolution begin to grow, and this would be favored within permissive regional “microenvironments” in the ovary. We describe this model of PFGA and subsequent patterns of follicle development and death as reminiscent of a ‘gauntlet,’ because follicles must overcome continuously changing regional stressors, and repair incurred damage in order to grow and survive.

In 2000, Finch & Kirkwood (2000) mentioned the possibility that ovarian follicle maturation might be dictated by “pure chance” in an extensive consideration of how randomness can influence key events during development and aging. Consideration of dynamic, regional changes in signals and cell stressors within the ovary over time led us to test whether ovarian aging could be described mathematically by a random process. RW modeling of PF behavior (similar to Diffusion Decision Modeling as described by Ratcliff & McKoon (2008), Ratcliff et al. (2016)) was found to recapitulate ovarian aging at the level of simulated subjects and across populations in terms of the ANM. This mechanistic RW determination of primordial follicle decay and ANM distribution is distinct from non-mechanistic approaches where a curve is constructed mathematically in order to fit a series of data points. Unlike curve fitting approaches (as assessed in Coxworth & Hawkes, 2010), the simulation approach used in this study generates synthetic data de novo which can be seen to recapitulate actual ovarian aging data. We next consider advantages and limitations of our use of RWs to model ovarian aging.

Advantages

The RW approach is quite simple. Our model in its simplest form was able to fit both cross-sectional follicle decay data and an ANM distribution by choosing only two free parameters. The pattern of PFGA can be considered a by-product consequence of physiologically fluctuating ISR activity relative to a growth threshold, without the need for complex signaling or sensing that dictates whether an individual PF begins to grow or stays dormant. It may be that a more complex mechanism(s) ultimately dictates this decision, but the RW process appears capable of giving rise to the natural pattern of follicle loss.

Importantly, this interpretation in no way contradicts existing data on genetic and environmental factors that influence the overall rate of PFGA. Evaluation of genetic model systems and large-scale Genome Wide Association Studies (GWAS) have revealed genes and pathways that influence the rate of ovarian aging and the ANM (Stolk et al., 2012; Day et al., 2015). For example, Antimüllerian hormone (AMH; Durlinger et al., 1999; Pankhurst, 2017), and Phosphatase and tensin homolog deleted on chromosome 10 (PTEN; Reddy et al., 2008; Jagarlamudi et al., 2009; Cheng et al., 2015) negatively regulate PFGA as seen in accelerated loss of the PF reserve in mouse knockouts. Loss of Tumor necrosis factor alpha (Tnfα; Cui et al. (2011)), or, Tumor necrosis factor receptor 2 (Tnfr2; Greenfeld et al. (2007)) has the opposite effect, where PFGA is significantly slowed and the duration of ovarian function is extended. Downstream NFκB signaling also has been implicated in ovarian aging (Stolk et al., 2012), and disruption of NFκB inhibitory proteins IκBα and IκBβ also significantly slows the rate of PFGA (Wright et al., 2020). In addition to key protein signals, micro RNAs (miRNAs) and longer noncoding RNA species (ncRNAs) have been shown to regulate PFGA (Yan et al., 2019; Zheng et al., 2019; Maalouf, Liu & Pate, 2016; Wang et al., 2016; Yang et al., 2013). In the case of ncRNAs let-7/H19, this occurs due to their regulation of AMH levels (Qin et al., 2019). It has not been clear, however, how these and other signaling pathways combine in their action in order to determine known patterns of PF loss (Erickson, 2008; Kallen, Polotsky & Johnson, 2018; Zhang & Liu, 2015; Reddy, Zheng & Liu, 2010; Ernst et al., 2017). We now hypothesize that it is the combined action of all such identified factors that ultimately influence ISR activity, and that this integration of stress, damage, and signaling events gives rise to patterns of ovarian aging.

The recapitulation of ovarian aging patterns by RWs also illuminates several features of the biological process that are fascinating but have been difficult to explain. First, it is well-known that growing (preantral) follicles are present during prepubertal life in the human ovary, but insight as to what controls their numbers has been lacking. The RW approach provides a reasonable explanation for the onset and continuation of PFGA prior to puberty, and also the characteristic “plateau” in PF numbers during this time. This is because, if the RW process begins around the time of birth, there is a delay before the first follicles can engage in RWs for enough time such that they can cross the PFGA threshold in appreciable numbers. As presented here, appreciable numbers of PFs begin to be available in the window of time corresponding to menarche onset. Next, the supply of growing follicles between puberty and late in the third decade of life is consistent. The RW model gives rise to this stable supply of PFs in this window of (simulated) time, given the application of drift. It is also well-known that PFs and occasionally, growing follicles are present in the postmenopausal ovary. The RW model favors the continued presence and decay of PFs after menopause, and given a reasonably stable rate of PFGA, the rare commitment to growth would still occur in these later years. Mechanisms that control the maximum ANM can also be considered in new ways. Because some simulated subjects have nearly an order of magnitude greater starting supply of PFs, one or more mechanism must be in place that limits the ANM at approximately 62 years. Following along the results presented here, either a high PF starting supply must correspond to accelerated follicle loss compared to lower PF starting supplies, or the loss of the PFGA-slowing action of factors like AMH leads to acceleration in PFGA such that women almost never have more than 1,000 PFs remaining by age 62. These possibilities can be modeled as alterations in drift.

Our provision of the formal mathematical treatment of continuous and discrete-step models, and also user-modifiable code for discrete-step modeling makes it possible for interested parties to both reproduce our results, and also to test the effects of modifying the models upon patterns of PFGA and ovarian aging. Users can “tune” their modeling conditions as we did in cases of starting supply, drift, and diffusivity, with additional conditions that may better reflect actual signaling within the ovary, including mechanisms that have yet to be identified. Especially in the case of the discrete step RW R code, tuning can be performed by non-mathematics professionals. A final advantage of such tuning is evident in the way that inter-subject variability relates to patterns of ovarian aging and population-level ANM. While our results support starting supply as a plausible central factor in determining how the ANM differs between women (consistent with Depmann et al. (2015)), modeling output is improved when both starting supply and Subject-Variable drift are applied. The RW approach can be refined (and compared to alternative approaches) as biological mechanisms are revealed and higher quality female reproductive aging datasets become available.

Limitations

Despite what we consider compelling advantages, our approach has several limitations. First, we must acknowledge that the decades-long time scale of the process makes the normal behavior of individual PFs very difficult to track for direct follicle-by-follicle comparison to modeling results. In terms of the RW approach, while it is true that we only needed two free parameters to fit available follicle decay data and the known human ANM distribution, we acknowledge that there are many possible parameters (e.g., degrees of freedom) that could be tuned in order to produce RWs that faithfully reproduce these natural patterns. We also acknowledge that there may be differences between primordial follicles that influence their individual likelihood of growth activation in a more deterministic fashion. However, the RW model tuning that we performed was justifiable in terms of known biological measurements (starting supply) and the action of biological signaling factors (as in the modeling of accelerating PFGA due to declining AMH as the stepwise alteration of RW drift). It is true that the mathematical treatment of those biological features was done with the defined goal in mind of matching patterns of ovarian aging. Even so, it is satisfying that the action of a RW can so closely match PF loss and the ANM distribution when constructed in a logical fashion and subject to so few model variables.

We also acknowledge limitations with the best-available datasets used for RW modeling. As the authors that compiled the dataset(s) of histological PF numbers over time noted, how ovarian specimens were procured may influence the evaluated pattern of PF decline (Wallace & Kelsey, 2010). It is reasonable to expect that ovaries collected during autopsy are mostly representative of a random sampling from the general population. However, especially at later ages, ovary removal during elective surgery is less likely to be representative of the general population. Until non-invasive and accurate methods of PF number estimation become available, we and others will remain limited to cross-sectional data of PF numbers such as these. There are also caveats related to available datasets for the human ANM distribution. These are related to imprecision due to patient self-reporting their timing of menopause (defined as a year without menses; Weinstein et al., 2003; McKinlay, Bifano & McKinlay, 1985; McKinlay, Brambilla & Posner, 1992), and also to the fixed, artificial threshold of 1,000 PFs (Faddy & Gosden, 1995) used to define menopause onset in simulated women.

Next, despite expression of core ISR machinery in pregranulosa and granulosa cells, we have limited direct experimental evidence that ISR activity varies relative to a threshold in the pregranulosa cells of human PFs in vivo. We have generated immunofluorescence data that shows variable levels of the core ISR regulatory factor P-eIF2α in mouse PF pregranulosa cells and oocytes (Hagen-Lillevik, submitted, preprint available at https://www.researchsquare.com/article/rs-1682172/v1). Even so, despite our prior detection of P-eIF2α in human nongrowing follicles (Llerena Cari et al., 2021), a study comparing P-eIF2α levels in nongrowing follicles in multiple replicate human ovary specimens has not yet been performed. While it is highly likely that regional differences in stress and damage occur similarly between the mouse and human ovary, it may be that mechanisms that respond to these dynamic local conditions differ between the species. As mentioned, stochastic ISR checkpoint resolution is our favored model of the switch that activates PFGA, but the proposed RW could be greatly influenced by signals that function separately, or parallel, to the ISR.

Final considerations

RWs appear to provide a useful “systems biology” (Kitano, 2002) framework for the modeling of human ovarian aging. We consider the following final points the major implications of our approach. First and foremost, the recapitulation of ovarian aging by RWs suggests that random action may be an evolutionary strategy used to ensure that a minimum duration of ovarian function occurs in almost all women. It may be that more strict, deterministic PFGA control mechanisms would be more susceptible to dysregulation, and that randomness is protective against catastrophic PF loss. Next, our main model variables, PF starting supply and RW drift, are likely to be determined both by genetic/inherited factors in individuals and by environmental exposures. Genes identified in loss-of-function experimental studies and GWAS approaches may influence starting supply and/or drift. In the case of drift, these would likely be genes involved in the resolution of stress and cellular damage as at least in our model, these would modulate the probability of checkpoint resolution and cell cycle entry. Similarly, exposure to factors like chemotherapeutic agents could be acting at the level of drift, as the acute cellular response to damage would be upregulation of repair factors that again should impact the probability of checkpoint resolution, cell cycle entry, or death. Future experiments may be able to address these questions directly.

In the current model we treat the decision to undergo PFGA as if a single unit enters the cell cycle. As more information becomes available, we can more precisely model the multiple pregranulosa cells in human PFs, and how interactions between pregranulosa cells and with the oocyte influence RW behavior. We do not suggest that individual PFs behave in unpredictable ways. Instead, PFs likely respond to their local conditions in regulated, nonrandom ways, as dictated by known signaling pathways. PFs are exposed to conditions that change over time within the ovary, and simple, random processes are shown here to closely recapitulate the pattern of PFGA over time. We can therefore consider the mammalian ovary as a non-trivial, emergent, self-organizing system, with limited adaptive capacity. The ovary’s critical functions as the transmitting organ of the female germline as well as its support of female health and well-being prior to menopause are clearly non-trivial. That the simple system can deliver a consistent supply of maturing follicles that meets reproductive and endocrinological needs for decades is suggestive of emergent self-organization. Last, the ovary’s adaptive capacity is limited by the requirement that eggs produced can support reproduction by that subject. Adaptation can thus occur that alters ovarian function, but not in a way that compromises reproductive potential. Diverse future approaches including direct experimentation and further model refinement can be used to investigate how the pattern of ovarian aging in individual subjects and across populations of mammals, including women, is influenced by random behavior of ovarian follicles.

Supplemental Information

Supplemental Information 1 Starting supply distribution.

A) A histogram of the 30 PF counts for women near birth reported by Wallace & Kelsey (2010) (blue bars), which is well-approximated by the log-normal distribution in Eq. (8)–Eq. (9) (dashed black curve). B) A cumulative distribution function plot of observed PF counts (blue solid line) vs. the log-normal distribution (dashed black curve).

Click here for additional data file.

Supplemental Information 2 ANM histograms generated from RW output when drift was set as specified in ”Drift Parameters” column.

As shown, an ANM distribution centered around a median age of approximately 51 (red vertical line) can be produced in each case, with few simulated subjects reaching menopause before 40 years and after 60 years. Time-Variant drift indicated by the asterisk (*) was applied by modifying drift conditions and also applying a single step drift acceleration in year 38 of simulation time. This was used to interrogate the possibility that PF loss accelerates during reproductive aging. Note that here, the ANM distribution generated when Subject-Variable drift is applied (middle panel) is broader than that seen for homogenous drift (top panel) given otherwise identical model conditions. Application of Time-Variant drift resulted again in a narrower ANM distribution, and prevented simulated subjects from reaching the ANM threshold after age 62 (blue vertical line).

Click here for additional data file.

Supplemental Information 3 Appendix: Supplemental treatment of discrete and continuous random walk models and interrogation of the human starting supply distribution of primordial ovarian follicles.

We first expand upon the alignment between continuous and discrete random walk models, then provide the justification for treatment of PF starting supply as a log-normal distribution (Fig. S1), and finally provide ANM simulation results as histograms for further consideration (Fig. S2).

Click here for additional data file.

Nanette Santoro, M.D., Amanda Kallen, M.D., Aaron Clauset, Ph.D., and Kelle Moley, M.D. are gratefully acknowledged for key suggestions provided during the development of the manuscript. The graphical abstract was produced by Kimen Design4Research.

Additional Information and Declarations

Competing Interests

Author Contributions

Data Availability

The authors declare that they have no competing interests.

Joshua Johnson conceived and designed the experiments, performed the experiments, analyzed the data, prepared figures and/or tables, authored or reviewed drafts of the article, and approved the final draft.

John W. Emerson conceived and designed the experiments, performed the experiments, analyzed the data, prepared figures and/or tables, authored or reviewed drafts of the article, and approved the final draft.

Sean D. Lawley conceived and designed the experiments, performed the experiments, analyzed the data, prepared figures and/or tables, authored or reviewed drafts of the article, and approved the final draft.

The following information was supplied regarding data availability:

The MATLAB code is available at figshare: Johnson, Joshua (2022): Matlab files. figshare. Software. DOI 10.6084/m9.figshare.19834774.v1.

The data and R code are available at figshare: Johnson, Joshua (2022): R files and Primordial Follicle Dataset. figshare. Software. DOI 10.6084/m9.figshare.19858987.v1.

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
