# Peer review of "Recapitulating human ovarian aging using random walks"

_PeerJ, doi:10.7717/peerj.13941_

## Round 0.1 · original submission · Minor Revisions

As you can see, all reviewers are extremely enthusiastic about your work and recommended acceptance after some minor revision. Please address concerns of the reviewer #2 and make necessary amendments.

·

Basic reporting

This paper builds on the recent literature on the integrated stress response to investigate the hypothesis that checkpoint resolution allows/permits/triggers active cell cycle in the human ovary. Code and data are provided that show how random walk models based on ISR can be derived and used.


The maths and code are all fine. I'm still struggling to fully understand the concept and mechanisms of ISR, but (a) this not my area of expertise, and (b) the work is strongly supported by the cited literature.

Experimental design

The random walk approach is novel and interesting. Deriving estimates for the diffusivity and drift parameters from the literature on age-related primordial follicle population in healthy females is clearly a sensible approach.

Validity of the findings

The results are in agreement with other recent studies in rodent, large mammal and primate models that, taken together, suggest that that changes in mechanical stress in the extra-cellular matrix and granulosa cells strongly affect primordial follicle activation and growth of good quality oocytes. [Nagamatsu et al. Sci Adv. 2019;5(6);
Devos M et al. Follicle Activation by Physical Methods and Clinical Applications. In: Grynberg M, Patrizio P, eds. Female and Male Fertility Preservation;
Xu M et al. Biology of reproduction. 2006;75(6);
West ER et al. Biomaterials. 2007;28(30);
Hornick JE, et al. Human reproduction. 2012;27(6)]

It is interesting to see that the old controversy of a sharp change in population decline at age 38 arises again under the new (and forward-looking) random walk approach. Faddy & Gosden reported a bi-exponential population model with a step change at age 38 as the best fit to the histological data available. In the absence of any known event that occurs in human females at about 38 years of age, other researchers preferred smoother (and hence more complex) models. As noted here, the differences are small, especially when the wide ranges for population at any given age are taken into account. The key point is that not finding and reporting such a step change would have been strange, when seeking models with the smallest feasible number of parameters.

Demonstration that the results are close to a widely accepted growth/decline model from conception to menopausal years supports the approach taken. In fact, the RW models shown in Fig 2a & b are potentially an improvement on the Wallace Kelsey model used for comparison: the plateau for prepubertal ages is more pronounced for RW, which is biologically more plausible. Of course both approaches are to an extent validated by inferring menopausal ages very close to those empirically observed in population studies, without these ages forming part of the model characteristics.

Additional comments

Fig 3 (and the underpinning code) is potentially very useful for oncofertility researchers. Until now, a simple modelling assumption has been posited (but never validated) fro the rate of decline in follicle population after a cytotoxic insult received during cancer treatment. The assumption is that the treatment has aged the patient in terms of ovarian reserve, and that her decline will now continue as for a healthy women of the new and older age. This assumption is shown not to be windy incorrect in Fig 3, but crucially it can now be tested in the light of information that could inform the estimation of RW parameters. On reflection, this also holds for the healthy case: the code allows staring supply to be augmented with drift variability, allowing the empirical testing of another key but as yet unvalidated assumption in previous models, namely that a large/small starting supply is linked to a late/early menopause.

Overall, the RW contribution reported here is significant improvement on Wallace-Kelsey, Faddy Gosden, etc. as it (i) is in broad agreement with expected aspects of age-related follicle populations dynamics, (ii) is forward looking (rather than reporting a regularised best fit to a static dataset), (iii) is more closely related to the stresses and internal structure of the ovary, and (iv) allows further detailed investigation of poorly understood aspects by augmenting the peak model with variable drift.

Reviewer 2 ·

Basic reporting

The manuscript "Recapitulating human ovarian aging using random walks"
by Johnson, Emerson, and Lawley describes a very simple model for the
aging of ovarian follicles. Each follicle is described as undergoing a
random walk with drift towards a defunct state. As more and more
follicle eventually age out, females lose fecundity. Although the
basic numbers used are realistic, the proposed representation is meant
to be a toy model that captures the basic features. There really isn't
a discriminating fitting with data so it is unclear if any number of
deterministic model (but with heterogeneity) could yield similar
results. For example, could the evolution of each follicle be
considered deterministic (possibly with aging) but with a distribution
of initial states? Could an appropriate distribution convect towards
death and yield similar results?

Overall, the model and analysis appears technically correct, although
I am not sure how much new insight is offered. The writing is decent,
but appears patched together in some places. There is also some
unconventional exposition like capitalizing Starting Supply, Drift,
and Diffusivity (?). Each author should read through and edit
serially.

Experimental design

Okay

Validity of the findings

Appropriate. Model is self-consistent and data are shown.

Reviewer 3 ·

Basic reporting

No comment.

Experimental design

No comment.

Validity of the findings

No comment.

Additional comments

This is one of the most interesting, innovative and clearly written manuscripts this reviewer has seen in quite a long time. The authors have created a mathematical model of human ovarian aging, i.e. longitudinally documenting the loss of primordial follicles from the original population present at birth throughout 5-6 subsequent decades. The authors assume that primordial follicles receive signals according to their local environment within the ovarian cortex, and that the pattern of follicle activation over time will reflect a one-dimensional random walk that includes a growth threshold. The authors provide a basis for the regional importance based on their prior work in mice wherein they demonstrated that the Integrated Stress Response (IRS) changes over time within regions of the ovary and would thus generate patterns based on follicle growth and dormancy due to the regional stressors and repair of damage that would reflect follicle loss and survival.
This reviewer has no individual or specific comments that need to be addressed in the manuscript, it is that well-written. Rather, comments on the strengths of the manuscript are offered below:
No one in the field of ovarian biology has attempted to create/perform such a mathematical model; most data on human primordial follicle loss with age has been gleaned from historical records of histological specimens and follicle counts that may or may not reflect accurate measurements of follicle loss because they do not include what is happening in an entire ovary, but rather just a few representative sections. Thus, the approach taken in the present manuscript is highly innovative as well as informative.
Additional strengths of the authors’ model include not only the growth threshold as a critical parameter, but accounts for population heterogeneity as well as comparisons of the rate of follicle loss based on the numbers of primordial follicles in a given woman’s reserve at birth that would represent follicle loss influenced by genetics or environmental exposures, in addition to acute decreases in follicle numbers at different ages across the lifespan as would be seen in female cancer patients receiving gonadotoxic cancer treatments. A further strength is development of user-friendly R code so that other investigators can interrogate follicle loss by altering other conditions of specific importance and/or adding variables based on their own experimental data.
It is truly remarkable that the resulting graphs of loss of follicle numbers over the lifespan calculated by the Random Walk model reflect so very closely that of a model previously proposed by Kelsey and colleagues. Perhaps this is due to the fact that the authors based the starting supply of follicles on the Kelsey data. However, this cannot be the entire explanation because the authors the utilized Random Walk model and included additional considerations (growth threshold and population heterogeneity) not present in the Kelsey model. Adjusting just these two important variables led to a physiologically believable model.
The authors are careful to discuss the fact that there are many factors reported to activate or suppress primordial follicle growth, so many such that testing all of the possible combinations would be rather onerous and since the combined action of these factors is physiologically unknown at present. It is reasonable for the authors to assume that IRS activity integrates these various signals, and this was a great place to start in the creation of the RW model.
Several limitations with the authors’ model were clearly presented and very inclusive and discussion of these parameters are greatly appreciated by this reviewer. The authors are careful to point out that ISR pathways have been documented in mice, but data is lacking that documents changes in ISR activity in individual nongrowing human follicles. Nonetheless, their reasoning that changes in the local environment of a given primordial follicle would occur in both species and dynamic responses must occur to regulate whether a follicle grows or remains dormant is again, a reasonable foundation for building their model.
Lastly, the authors provide some provocative ‘food for thought’ that has not yet been presented in the literature on primordial follicle loss with age. The authors propose the concept that the ovary is a self-organizing system with limited adaptive capacity that is necessary for the consistent supply of follicles and hormones over decades as an evolutionary strategy and that random action protects against catastrophic loss of follicles to help insure maintenance of reproductive potential; what an interesting thought!
This is an outstanding manuscript that represents an innovative way to look at primordial follicle loss/activation that can serve as a basis for further testing by other investigators. Kudos to the authors for a well-organized, thoughtful and diplomatic discussion. I hope this critical analysis and model development does not get lost in the myriad of publications such that practitioners of ovarian biology research cannot find it among their ‘usual’ journals and that it will be picked up for a commentary with wider exposure.

---

## Round 0.2 · accepted · Accept

Concerns of the reviewer were adequately addressed and the manuscript was amended accordingly. Therefore, I am please to accept this work in its present form.